# Loess Strata Distribution Characteristics and Paleoclimate Spatial Pattern during the Last Interglacial in the Luohe River Basin

Jiao Guo [1,2], Jiansheng Shi [3], Hongyun Chen [1,2], Chao Song [1,2], Qiuyao Dong [1,2] and Wei Wang [1,2,*]

1   Institute of Hydrogeology and Environmental Geology, Chinese Academy of Geological Sciences, Shijiazhuang 050061, China; guojiao@mail.cgs.gov.cn (J.G.); chenhongyun@mail.cgs.gov.cn (H.C.); songchao@mail.cgs.gov.cn (C.S.); dongqiuyao@mail.cgs.gov.cn (Q.D.)
2   Key Laboratory of Quaternary Chronology and Hydrological Environment Evolution, China Geological Survey, Shijiazhuang 050061, China
3   Aero Geophysical Survey and Remote Sensing Center for Natural Resources, China Geological Survey, Beijing 100083, China; shijiansheng@mail.cgs.gov.cn
*   Correspondence: ww@mail.cgs.gov.cn

**Abstract:** Paleoclimate studies of loess in China have focused mostly on the time series of a single borehole or profile. However, research on loess strata and regional paleoenvironmental patterns could facilitate a deeper understanding of loess as a paleoenvironmental indicator and provide new insights into interpreting loess in sedimentary records of the paleoclimate. In this study, we determined the spatial pattern and regional characteristics of the paleoclimate during the Last Interglacial period in the Luohe River Basin, Shaanxi Province, China. We selected four representative boreholes in the study area (ZK04, ZK18, ZK13, and ZK19) from different landforms and zones, distributed from the northwest to the southeast, as well as three classic profiles (JB, JD, and LC). From north to south, comparative analyses were conducted of the loess strata, magnetic susceptibility, and grain size, and we analyzed the distribution characteristics of loess and paleosols in different geomorphological regions. The results showed that both the thickness and the sedimentation rate of loess in this river basin decreased from north to south. There were few paleosol horizons in the northern Liangmao area, but numerous such horizons in the southern plateau, and the degree of paleosol development increased from north to south. The magnetic susceptibility increased, whereas the particle composition tended to become thinner from north to south. The climate fluctuations of the Last Glacial recorded by the loess and paleosols in different regions were inconsistent.

**Keywords:** loess; paleosol; magnetic susceptibility; grain size; paleoclimate; spatial pattern

## 1. Introduction

Loess is considered one of the three pillars for studying past climate changes because of its continuous deposition, long chronological sequence, and easy access, with the other two pillars being deep-sea sedimentary sequences and polar ice cores [1]. The loess and paleosol sequences provide an excellent sedimentary record for understanding the evolutionary history of paleoclimate on the regional orbital and tectonic scales and also provide an important reference for understanding the changing trend on the suborbital scale [2]. For many years, researchers have conducted extremely systematic research on loess in Europe and China, while research on loess in Central Asia is relatively weak, with a focus on sedimentology and chronology of the late Quaternary loess strata [3]. The research shows that most of the loess in Central Europe is deposited in the periglacial zone between the Scandinavian ice sheet and the Alpine glaciers, while the loess in Central Asia and China is closely related to the surrounding gobi, desert, and mountain front deposits [4]. As typical aeolian sediments, loess has unique paleoclimatic value, from which local or regional environmental change information can be extracted. The Chinese loess can record

the evolution of the winter monsoon and East Asian summer monsoon controlled by the Siberian Mongolian high pressure [5]. During the interglacial period, the summer monsoon played a dominant role, forming paleosol under its temperature and humidity conditions. During the glacial period, it was dominated by the winter monsoon, leading to thicker loess deposits. The European loess and paleosol sequences have important research value for understanding the evolutionary history of westerly circulation in the westerly control area. However, the magnetic parameters of the loess in Europe indicate that the paleoenvironment and paleoclimate are different, and the paleoenvironmental significance is different in different regions. The magnetic susceptibility of most loess and paleosol sequences in Eastern Europe, located at the western end of the Eurasian loess belt, has a quasi-positive correlation with soil forming intensity [6], such as loess studies in Ukraine [7], Czech Republic [8], Hungary [9], Croatia [10], Romania [11], and Serbia [12]. In Siberia and Alaska [13], magnetic characteristics are negatively correlated with pedogenesis. However, the climate and topographic conditions in different regions of Central Asia are quite different, the loess sources are complex, and the paleoclimate and environment have certain uncertainties. The accumulation of loess in Central Asia and the development of paleosol in the atypical monsoon region are partly or indirectly affected by the Siberian Mongolian high pressure, as well as by climate changes in the North Atlantic and Mediterranean regions [4].

Studying the spatial patterns of the paleoclimate during specific cold and warm periods in the Quaternary is key to understanding the characteristics of atmospheric circulation, vegetation succession patterns, and the shifting range of climate zones during such periods. In the past, paleoclimatic studies of the Chinese Loess Plateau focused mostly on the time series of a single borehole or profile [14–18], with less research being conducted on spatial patterns, which were mostly studies of the spatial variation in loess grain sizes. In the early 1950s, Xiong and Wen [19] analyzed the particle size composition of surface soil in the loess regions of northern Shaanxi and Longdong, observing that particle size became finer from northwest to southeast. Subsequently, Liu et al. [20] and Liu [21] conducted field investigations and laboratory analyses of ten large profiles (six vertical and four horizontal) in the loess of the middle reaches of the Yellow River. Their research confirmed the finding that loess grain size gradually becomes finer from northwest to southeast. Moreover, the deposits could be divided into sandy loess, loess, and clay loess belts. Yang and Ding [22] conducted a detailed investigation and particle size analysis of 57 profiles above $S_2$ in the Loess Plateau, with the results showing that the particle size of both loess and paleosols was finer from north to south. Further, the particle size contours generally spread in an east–west direction, reflecting that the spatial differentiation of granularity was mainly north–south. Qiang et al. [23] conducted comparative research of the Luochuan and Weinan profiles along the gradient of climate zone change. These authors investigated the characteristics of the spatial variation of magnetic mineral content, mineral combinations, and magnetic domain size of loess–paleosol sequences in environments with different climate gradients in the Loess Plateau, as well as their paleoclimatic significance. Numerous researchers have used indicators such as magnetic susceptibility [24], the Rb/Sr value [25,26], free iron/total iron value [27], chromaticity [28], and soil micromorphology [29,30] to determine spatial variations of the strength of loess weathered into the soil. Zhao et al. [31] investigated the differences in paleodust fluxes in arid and semi-arid regions of China and their indicative climatic significance. Several scholars [32–36] conducted quantitative or semi-quantitative reconstruction and integration of indicators such as paleotemperature, paleoprecipitation, and other factors in characteristic periods. In these investigations, published information was considered, as well as climate models applied to simulate the environmental patterns of the paleoclimate under instantaneous or equilibrium conditions (including high and low air pressure, rainfall, and temperature) and analyze the factors possibly influencing the climate and the mechanisms of climate change.

The Luohe River Basin in Shaanxi is located in the middle of the Loess Plateau and is an area of typical eolian loess deposits. The sedimentary loess–paleosol sequence is

continuous and complete, with good spatially comparable climate records. This area is considered arid and semi-arid, is spatially distributed from northwest to southeast and to the north and west, and is adjacent to the inland desert in Asia. Further, this area is extremely sensitive to climate change and is affected significantly by the East Asian monsoon. However, the influences of the monsoon on the northwestern and southeastern margins of the basin differ significantly, with the southeastern margin being affected considerably by the East Asian summer monsoon. Conversely, the northwestern region is close to the desert and neighboring belts and, as it is located at the edge of the Asian monsoon region, is affected more by the winter monsoon. Therefore, the basin region is an excellent research area and vector for studying regional paleoenvironmental patterns. Ample research has been conducted on Quaternary loess in the Luohe River Basin, with the results providing important stratigraphic information and data support for this current study. For example, the research includes the JB [37], JD [38], and LC profiles [39] (which are the most detailed profiles). The profiles are typical of loess strata, and systematic comparisons have been conducted between these profiles and deep-sea isotope studies.

Different types of loess landforms are distributed in the Luohe River Basin from northwest to southeast, such as platforms, plateaus, and hills and ridges. To consider the influence of topographic and geomorphic factors on the regional paleoclimatic pattern and analyze the distribution characteristics of loess and paleosols in different geomorphological regions, we compared the stratum, magnetic susceptibility, and grain size of four representative boreholes with three classic loess profiles. The boreholes are ZK04, ZK18, ZK13, and ZK19, and the profiles are JB, JD, and LC (Figure 1). This comparison revealed the spatial pattern and regional characteristics of the paleoclimate during the Last Interglacial period in the Luohe River Basin.

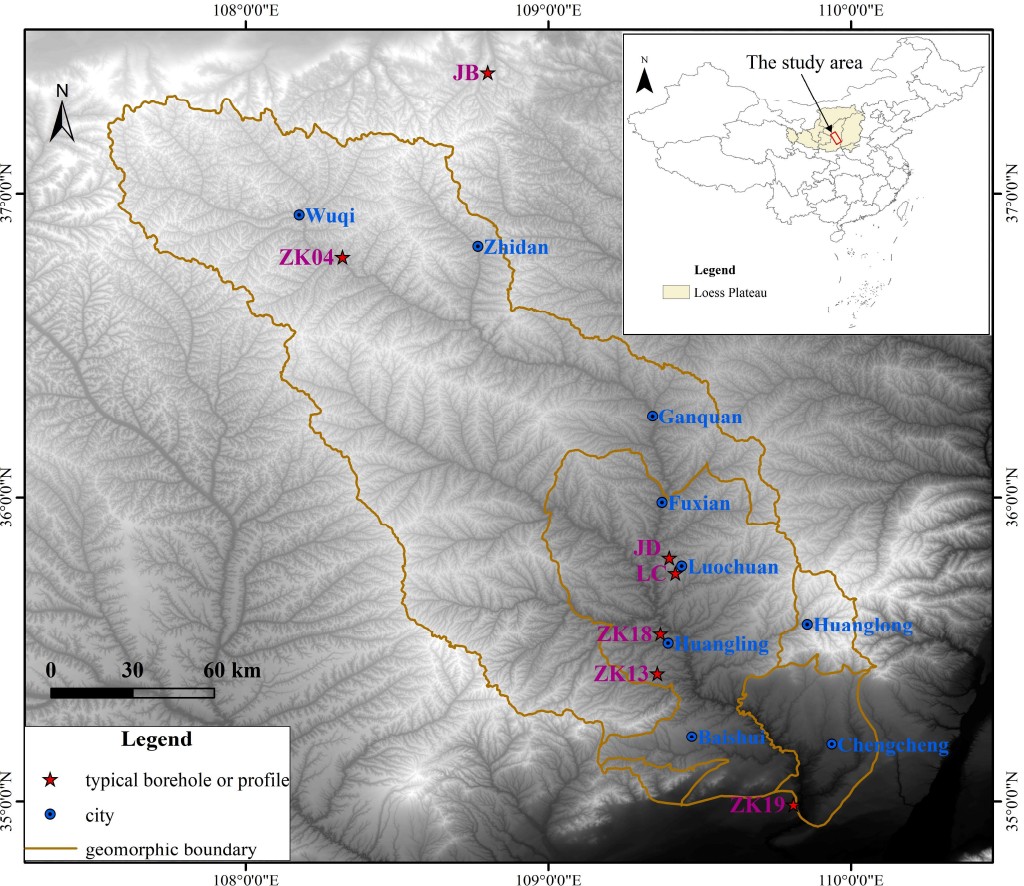

**Figure 1.** Map showing the distribution of main boreholes and classic profiles in the Luohe River Basin.

In the course of this study, we accumulated a certain amount of basic data for studying loess strata and regional paleoenvironmental patterns in China. These data facilitate further understanding of loess as a paleoenvironmental indicator and provide new insights into the interpretation of loess in sedimentary records of the paleoclimate.

## 2. Overview of the Study Area

The Luohe (or Beiluo) River, called Luoshui (or Beiluoshui) in ancient times, is the secondary tributary of the Yellow River and the primary tributary of the Weihe River. The Luohe River is the longest river in Shaanxi at 680.3 km in length. It originates from the Caoliang Mountain at the southern foot of the Baiyu Mountain, flows into the Weihe River from northwest to southeast, and passes through two terrains, namely the Loess Plateau and the Guanzhong Plain. In this study on the Luohe River Basin, the area where the Luo River flows into the Wei River Valley was not included. We only considered the Luo River Basin to the north of Yongfeng Town, between 107°32′24″–110°06′06″ longitude and 34°54′00″–37°19′48″ latitude. The river extends to the Baiyu Mountain in the north, Chengcheng and Pucheng counties in the south, Ziwu Ridge in the west, and Huanglong Mountain in the east, covering an area of 26,264 km$^2$. The general shape of the research area is that of a narrow, long belt obliquely crossing the Loess Plateau in northern Shaanxi from northwest to southeast. The terrain slopes from northwest to southeast, with the highest peak in the northwest reaching 1850 m, whereas the elevation of the valley area in Chengcheng County in the southeast is approximately 420 m. The difference in elevation of the entire watershed exceeds 1400 m. Numerous ravines occur in the watershed, and the average height difference of the slope from top to bottom is approximately 200–400 m.

From upstream to downstream, the entire Luo River Basin is covered with loess of varying thicknesses, except for part of the bedrock exposed at the bottom of a ditch. The loess is relatively thick (180–200 m) in the Luochuan area in the middle reaches of the Luohe River, whereas the loess thickness ranges from over ten meters to tens of meters in the upper reaches. In the Luohe River Basin, the loess is mainly from the Middle and Late Pleistocene, and multiple horizons of paleosols are observed.

## 3. Materials and Methods

### 3.1. Sample Collection

We collected information on the geographic location, stratigraphic characteristics of loess and paleosols, and magnetic susceptibility of the three classic profiles (JB [37], JD [38], and LC [39]) from the literature. Borehole cores from ZK04, ZK18, ZK13, and ZK19 were sampled at 2 cm intervals, obtaining a total of 4317, 7155, 6189, and 6228 samples, respectively.

### 3.2. Testing Methods

(1) Magnetic susceptibility tests were conducted on all the samples from boreholes ZK04, ZK18, ZK13, and ZK19. We weighed and ground 10 g of each sample until the soil particles were smaller than 2 mm. A Bartington® MS2 magnetic susceptibility meter (Bartington® Instruments Ltd., Witney, UK) was used for the measurements, and the low frequency (0.47 Hz) magnetic susceptibility ($\chi$lf) and high frequency (4.7 Hz) magnetic susceptibility ($\chi$hf) of each sample were measured at a suitable distance from the interfering magnetic field. Each sample was measured three times consecutively, with the average value calculated from these measurements.

(2) Particle size tests were conducted on 499,674, and 333 samples from the upper parts of the borehole cores of ZK04, ZK18, and ZK19, respectively. We weighed 2–4 g of each air-dried soil sample and placed the portions in a beaker. Subsequently, 10 mL of 10% $H_2O_2$ and 10 mL of 10% HCl were added to the beaker, and the mixture was heated on an electric heating plate until fully reacted and for the removal of organic matter and carbonates, respectively. After adding distilled water, the samples were left to stand for 24 h. The supernatant was extracted, and 10 mL of $(NaPO_3)_6$ dispersant at a

concentration of 0.05 mol/L was added to it. After ultrasonic vibration, the particle size frequency distribution was measured with a Malvern Mastersizer 2000 laser particle size analyzer (Malvern Panalytical, Malvern, UK), with a measurement range of 0.02–2000 μm. Particle size analysis was conducted to determine the percentage, median particle size, and average particle size of the various sediment components.

## 4. Results

### 4.1. Regional Structural Characteristics of Loess Strata

The stratigraphic sequence and structural characteristics of loess in China are influenced by the paleoclimate, paleogeomorphology, material source, transportation capacity, and pedogenesis process. Accordingly, differences occur in the sequences, structures, and thicknesses of loess strata in different geomorphological regions [40]. Considering the similarities and differences in the geomorphology and referring to the Map of Geomorphology Types of the Loess Plateau in China [41], the landforms of the Luohe River Basin were divided from north to south into a desert–loess transition zone, loess hills and ridges, loess plateau, and loess platform. Figure 2 and Table 1 show the characteristics of the stratum structure of loess boreholes and the representative profiles in different geomorphological regions.

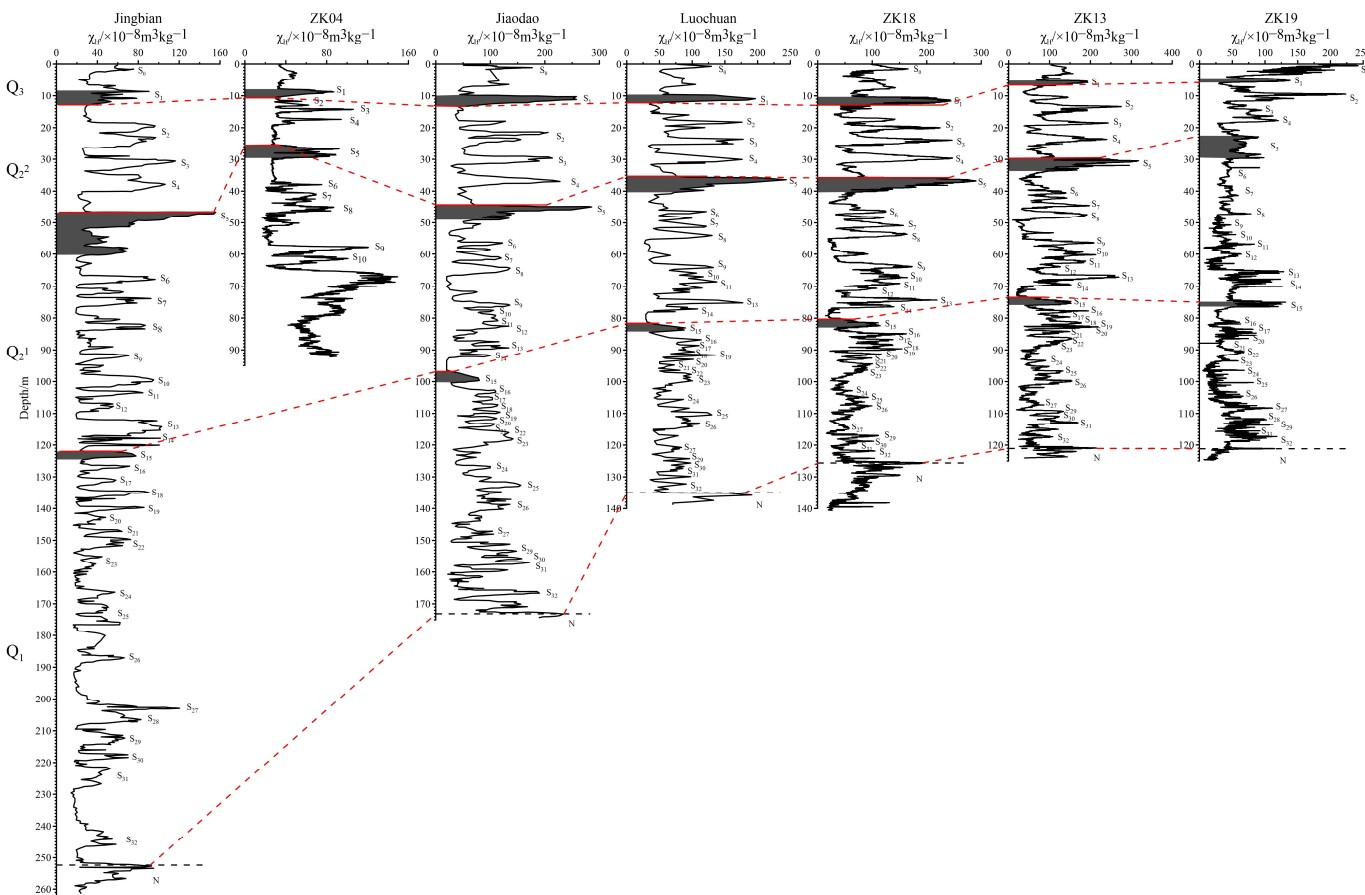

**Figure 2.** Comparison of the main boreholes and representative profiles in the Luohe River Basin.

**Table 1.** Thickness of loess stratum and characteristics of deposition rate of the main boreholes and representative profiles in the Luohe River Basin.

| Borehole or Profile | Landform Type | Location | Loess Thickness (m) | Deposition Rate (cm/ka) |
|---|---|---|---|---|
| JB | Desert–loess transition zone | 37.4° N, 108.8° E | 252 | 9.69 |
| ZK04 | Loess Hills and Ridges | 36.79° N, 108.32° E | 64.81 | 5.99 |
| JD | Loess Plateau | 35.8° N, 109.4° E | 173 | 6.65 |
| LC | Loess Plateau | 35.75° N, 109.42° E | 135 | 5.19 |
| ZK18 | Loess Plateau | 35.55° N, 109.37° E | 126 | 4.85 |
| ZK13 | Loess Plateau | 35.42° N, 109.36° E | 121 | 4.65 |
| ZK19 | Loess Platform | 34.99° N, 109.81° E | 121 | 4.65 |

### 4.1.1. Desert–Loess Transition Zone

JB profile (37.4° N, 108.8° E) [37] is located in the desert–loess transition zone in the north of the Luohe River Basin. This area is extremely sensitive to climate change. The total thickness of the loess profile is 252 m, containing 32 horizons of paleosols that are light in color. The black loess at the top of the profile is 1.1 m thick and covered with a thin horizon of loess.

### 4.1.2. Loess Hills and Ridges

ZK04 borehole (36.79° N, 108.32° E) is located in the northern area of the Luohe River Basin in Jinding Town, Zhidan County. The landform type of this area is typical hills and ridges of loess accumulation. The total thickness of the profile is 92.2 m, with the upper part containing 10 horizons of paleosols and the lower part containing lacustrine sediments. No black loess is exposed at the top of the profile, and the entire Quaternary is not uncovered.

### 4.1.3. Loess Plateau

JD profile (35.8° N, 109.4° E) [38] is located in the central part of the Loess Plateau, approximately 50 km north of the classic Luochuan profile. The loess profile is 173 m thick and contains 32 paleosols horizons. The black loess at the top of the profile is 1.2 m thick and is covered by thin loess horizons.

LC profile (35.75° N, 109.42° E) [39] belongs to a quasi-plateau subregion. The loess is 135 m thick and has 32 paleosol horizons. The black loess at the top of the profile is 1.0 m thick, with no loess covering.

ZK18 borehole (35.55° N, 109.37° E) is located in Tianzhuang Town, Huangling County, in the middle reaches of the Luo River Basin, where the landform type is a typical loess plateau remnant. The total thickness of the loess profile is 126 m, and it contains 32 horizons of paleosols with a deep color. The black loess at the top of the profile is 1.5 m thick, and there is no loess covering.

ZK13 borehole (35.42° N, 109.36° E) is located in Yaosheng Town, Yijun County, Tongchuan City, and the landform type is a loess plateau remnant. The loess profile is 121 m thick, contains 32 horizons of paleosols, and is dark in color. No black loess is exposed at the top of the profile.

### 4.1.4. Loess Platform

ZK19 borehole (34.99° N, 109.81° E) is located on the fourth-order river terrace of the Luohe River, and its landform type is a high fluvial platform. The loess profile is 121 m thick and contains 32 horizons of paleosols. The black loess horizon at the top of the profile is 1.0 m thick and is directly exposed at the surface without a loess covering.

In summary, the loess thickness in the Luohe River Basin varies significantly, including regional and local topographical differences. The loess thickness and the deposition rate both decrease from north to south. The loess at the northernmost JB profile is 252 m thick, with an average deposition rate of 9.69 cm/ka. The LC profile is 135 m thick, with an average deposition rate of 5.19 cm/ka. The loess thickness of the southernmost ZK19

borehole is 121 m, with an average deposition rate of 4.65 cm/ka. The ZK04 borehole was affected by the Wuqi Paleolake [42], with a different average deposition rate.

*4.2. Characteristics of Regional Paleosol Distribution*

Paleosols in loess can be divided into five groups in the vertical direction according to their lithology, structural development, color, and combination [43]; however, all five types might not be distinguishable in all loess accumulations. Differing geomorphic conditions in various geological and geographical areas, as well as factors such as the thickness of loess accumulation and the burial and retention of soil, could help preserve all five paleosol groups or one or more groups could be absent. The characteristics of the paleosol horizons of each group vary from region to region. Regardless of the color, thickness, degree of development, and number of paleosol horizons, the regional characteristics and patterns evidently vary.

4.2.1. Regional Distribution Characteristics of Black Loess ($S_0$) in Holocene Loess

As shown in Table 2, the black loamy soil in the Luohe River Basin is developed mainly in the southern plateau area and was preserved generally by being buried. The thickness is approximately 1.0 m, and this soil is covered with thin horizons of loess and one black loess horizon. However, in the northern hills and ridges area, black loam is often eroded or exposed only sporadically at the surface. Therefore, there was no or rarely exposed black loam in the hills and ridges.

**Table 2.** Thickness (m) characteristics of black loam and five paleosol horizons.

| Borehole or Profile | $S_0$ (Number of Horizons) | $S_1$ | $S_2$ (Number of Overlapping Horizons) | $S_3$ | $S_4$ | $S_5$ (Number of Overlapping Horizons) |
|---|---|---|---|---|---|---|
| JB | 1.1 (1) | 4.1 | 7.8 (2) | 3.5 | 4.8 | 13.3 (3) |
| ZK04 | \(0) | 1.9 | 1.5 (2) | 1.1 | 0.5 | 3.4 (3) |
| JD | 1.2 (1) | 2.8 | 3.2 (2) | 2.2 | 3.0 | 4.5 (3) |
| LC | 1.0 (1) | 2.3 | 4.6 (2) | 2.0 | 3.0 | 5.1 (3) |
| ZK18 | 1.5 (1) | 1.7 | 3.5 (2) | 1.2 | 2.1 | 4 (3) |
| ZK13 | \(0) | 1.3 | 2.2 (2) | 1.0 | 2.5 | 4.3 (3) |
| ZK19 | 1.0 (1) | 1.1 | 2.2 (2) | 0.7 | 1.9 | 6.9 (3) |

4.2.2. Regional Distribution Characteristics of Cinnamon-Type Paleosols

The first, second, third, and fourth horizons of paleosols from the black loam horizons are all cinnamon-type paleosols. The variations in the thickness of this group of paleosol horizons follow a regional distribution pattern. Table 2 shows the changes in the thicknesses of the five paleosol horizons from the surface to the bottom of the loess horizons and from north to south. The table also shows that the thicknesses of the first ($S_1$), third ($S_3$), and fourth paleosol horizons ($S_4$) follow a relatively evident pattern of thinning from north to south. The thickness of the second horizons ($S_2$) varies significantly owing to double-horizons stacking.

The soil development in the hills and ridges area in the north was weaker than that in the plateau area in the south. The paleosol development horizons in the ZK04 borehole were not quite clear, but those in ZK18 and ZK13 boreholes in the southern plateau area were clear.

The thickness of the loess horizons between these horizons of buried paleosols was generally 2–6 m, i.e., exceeding the thickness of the paleosol horizons, and it was roughly the same in the Luohe Basin area.

4.2.3. Characteristics of Regional Variations in Brown Earth Paleosols

Generally, the fifth horizon of paleosols ($S_5$) from the surface is a thick horizon of brown earth paleosol. The distribution of this horizon has regional stratigraphic significance. The variation in thickness of this paleosol horizon also follows a pattern of thickening from

north to south (Table 2). In the ZK04 borehole, the thickness of $S_5$ was 3.4 m, whereas that of $S_5$ increased to 5.1 and 6.9 m in the southern LC profile and the ZK19 borehole, respectively.

In some areas, the thickness of the fifth paleosol horizons in the loess horizons was less, similar to the thickness of the other paleosol horizons, or it did not exhibit three brief depositional breaks. This phenomenon is related to differences in the paleogeographical environment, depositional conditions, and regional climatic conditions in which the paleosols formed when the loess material was being deposited.

### 4.2.4. Characteristics of Regional Variations in Cinnamon-Type Deep-Buried Paleosols

Below the fifth paleosol horizons, the sixth ($S_6$), seventh, eighth, and ninth horizons, and even the eleventh and twelfth floors in some areas (and the fifteenth horizons in Luochuan, Shaanxi) were all cinnamon-type buried paleosols. The thickness of the sixth paleosol horizon ($S_6$) was small all over, and the distance between this horizon and the previous paleosol horizons above was generally large.

### 4.2.5. Characteristics of Regional Variations in Degraded Paleosols

The soil structure of this group of paleosols was evidently degraded, with a small thickness, dense horizons, and small intervals between each horizon (more than 10 or even 20 horizons). There were mostly dense, thin-horizons, degraded paleosols, with different horizons, below the eighth and ninth paleosol horizons in the Luohe River Basin.

Table 3 shows the differences in the distribution of the paleosol horizons in loess accumulations during various geological periods in the different regions of the Luohe River Basin. However, generally, a small number of paleosol horizons were developed in the hills and ridges in the north, whereas the paleosols in the southern plateau were the most developed at 20–30 horizons. The thickness of the loess accumulated in $Q_1$ and $Q_2$ tended to become thinner from north to south, whereas that of the loess in $Q_3$ did not change significantly.

**Table 3.** Distribution of paleosol horizons in loess accumulations during the Quaternary.

| Borehole or Profile | $Q_3$ | | $Q_2$ | | $Q_1$ | |
|---|---|---|---|---|---|---|
| | Number of Paleosol Horizons | Thickness/m | Number of Paleosol Horizons | Thickness/m | Number of Paleosol Horizons | Thickness/m |
| JB | 1 | 10.3 | 13 | 109.3 | 18 | 129.9 |
| ZK04 | 1 | 10.5 | 9 | \ | \ | \ |
| JD | 1 | 11.1 | 13 | 83.9 | 18 | 76.1 |
| LC | 1 | 10.6 | 13 | 69.9 | 18 | 53.0 |
| ZK18 | 1 | 10.9 | 13 | 67.3 | 18 | 45.8 |
| ZK13 | 1 | 6.5 | 13 | 67.5 | 18 | 47.0 |
| ZK19 | 1 | 4.8 | 13 | 69.3 | 18 | 45.9 |

### *4.3. Differences in Spatial Characteristics of Magnetic Susceptibility*

The mechanism of the increased magnetic susceptibility of paleosols remains controversial; however, pedogenesis is accepted widely as the cause [44,45]. As paleosols are a product of the natural landscapes in geological or historical periods, they record the influence of factors such as parent material, climate, biome, topography, and time on the surface materials at the time [46,47]. The spatiotemporal distribution, developmental form, and formation mechanism of paleosols in loess are restricted by the bioclimatic environment in the soil distribution zone as well as by the local topography and hydrology [48]. Therefore, the degree of paleosol development differs in various regions.

Although the magnetic susceptibility curves of loess sequences in various regions differ in shape, their trend remains consistent (Figure 2). The $S_5$ composite paleosol horizons had three obvious parts on each section, i.e., three overlapping paleosol horizons with three red strips, but the thickness varied from section to section. Apart from the ZK04

borehole, affected by the Wuqi Paleolake, and the ZK19 borehole, influenced by the river, the magnetic susceptibility value of $S_5$ was the highest in all the other profiles.

Figure 3 shows the characteristics of variations in the low-frequency magnetic susceptibility of ZK04, ZK18, and ZK19. Borehole ZK04, located in the northernmost part of the Luohe River Basin, had an average low-frequency magnetic susceptibility of $39.66 \times 10^{-8}$ m$^3$/kg in loess horizons $L_1$ and $71.04 \times 10^{-8}$ m$^3$/kg in paleosol $S_1$. In ZK18, located in the middle of the Luohe River Basin, the average low-frequency magnetic susceptibility was $95.43 \times 10^{-8}$ m$^3$/kg and $182.08 \times 10^{-8}$ m$^3$/kg in loess horizons $L_1$ and paleosol $S_1$, respectively. Borehole ZK19, located at the southernmost end of the Luohe River Basin, had an average low-frequency magnetic susceptibility of $100.35 \times 10^{-8}$ m$^3$/kg and $125.81 \times 10^{-8}$ m$^3$/kg in loess horizons $L_1$ and paleosol $S_1$, respectively. These characteristics indicated an increasing trend in the values of low-frequency magnetic susceptibility from north to south for both loess and paleosols. Moreover, owing to the influence of rivers in ZK19, the magnetic susceptibility of its paleosols was abnormal.

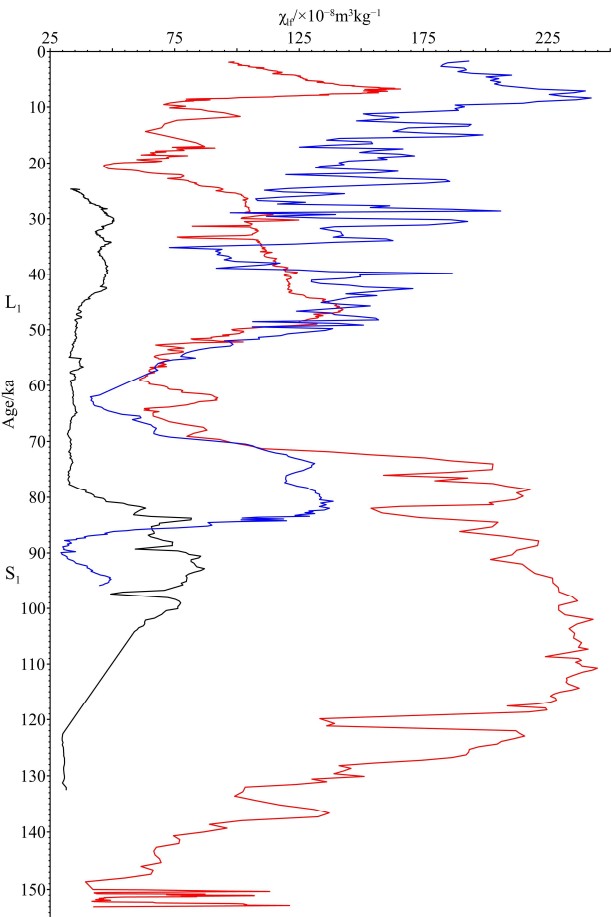

**Figure 3.** The characteristics of variations in low-frequency magnetic susceptibility of ZK04, ZK18, and ZK19. The black, red, and blue curves represent the magnetic susceptibility characteristics of ZK04, ZK18, and ZK19, respectively.

Table 4 shows that magnetic susceptibility increased from north to south, implying that the degree of paleosol development increased from north to south. The degree of development of the JB profile, located in the desert–loess transition zone, was not good despite numerous paleosol horizons being developed. Further, the value of magnetic susceptibility fluctuated between $14.3 \times 10^{-8}$ m$^3$/kg and $155.79 \times 10^{-8}$ m$^3$/kg). In the southern loess plateau, the value of magnetic susceptibility increased to approximately $300 \times 10^{-8}$ m$^3$/kg. However, owing to the influence of the coarse lithofacies and river

sediments, such as sand, sand pebbles, and pebble horizons in the ZK19 borehole, its magnetic susceptibility decreased.

**Table 4.** Characteristics of magnetic susceptibility of main boreholes and representative profiles in the Luohe River Basin.

| Borehole or Profile | Low-Frequency Magnetic Susceptibility ($\times 10^{-8}$ m$^3$/kg) |
| :---: | :---: |
| JB | 14.30–155.79 |
| ZK04 | 17.05–149.40 |
| JD | 20–286 |
| LC | 27–244 |
| ZK18 | 15–291 |
| ZK13 | 9.60–318.05 |
| ZK19 | 5.20–242.35 |

*4.4. Spatial Characteristics of Loess Grain Size since 0.13 Ma*

As detailed particle size measurement data were not available for the JB, JD, and LC profiles, only the particle size measurements of ZK04, ZK18, and ZK19 were used for analyzing the north–south variation in the grain size characteristics of the loess in 0.13 Ma.

Figure 4 shows three parameters of particle size, namely the percentages of <4 m (clay) particles and 4–63 m (silt) particles, and the median particle size d(0.5) [49]. As the figure indicates, in loess horizons $L_1$ of ZK04, which is located in the northernmost part of the Luohe River Basin, the average percentage of <4 m particles was 13.95%, the average percentage of 4–63 m particles was 86.05%, and the average median particle size was 28.35 m. In paleosol $S_1$, the average percentage of clay particles was 18.06%, that of silt particles was 81.84%, and the average median particle size was 24.06 m. For ZK18, located in the central region, the average percentage of <4 m particles was 22.12%, that of 4–63 m particles was 77.87%, and the average median particle size was 17.29 m in loess horizons $L_1$. In its paleosol $S_1$, the average percentage of clay particles was 28.31%, that of silt particles was 71.69%, and the average median particle size was 13.44 m. For loess horizons $L_1$ of ZK19, located in the southernmost part of the Luohe River Basin, the average percentage of <4 m particles was 29.97%, that of 4–63 m particles was 70.03%, and the average median diameter was 12.54 m. In paleosol $S_1$, the average percentage of clay particles was 30.52%, the percentage of silt particles was 69.48%, and the average median particle size was 11.16 m. These characteristics indicated that the dust particles had undergone a strong sorting process from north to south during their aerial transport. Several scholars regard the winter monsoon as the main transport force of Quaternary dust in the Loess Plateau [50–52], which is consistent with our conclusions from the information gleaned from the loess boreholes.

The results of particle size analyses showed that the particle compositions of both the loess and the paleosol horizons became finer from north to south, indicating significant north–south differentiation during the transportation process of airborne dust, with the coarse particles being deposited first and the fine particles later.

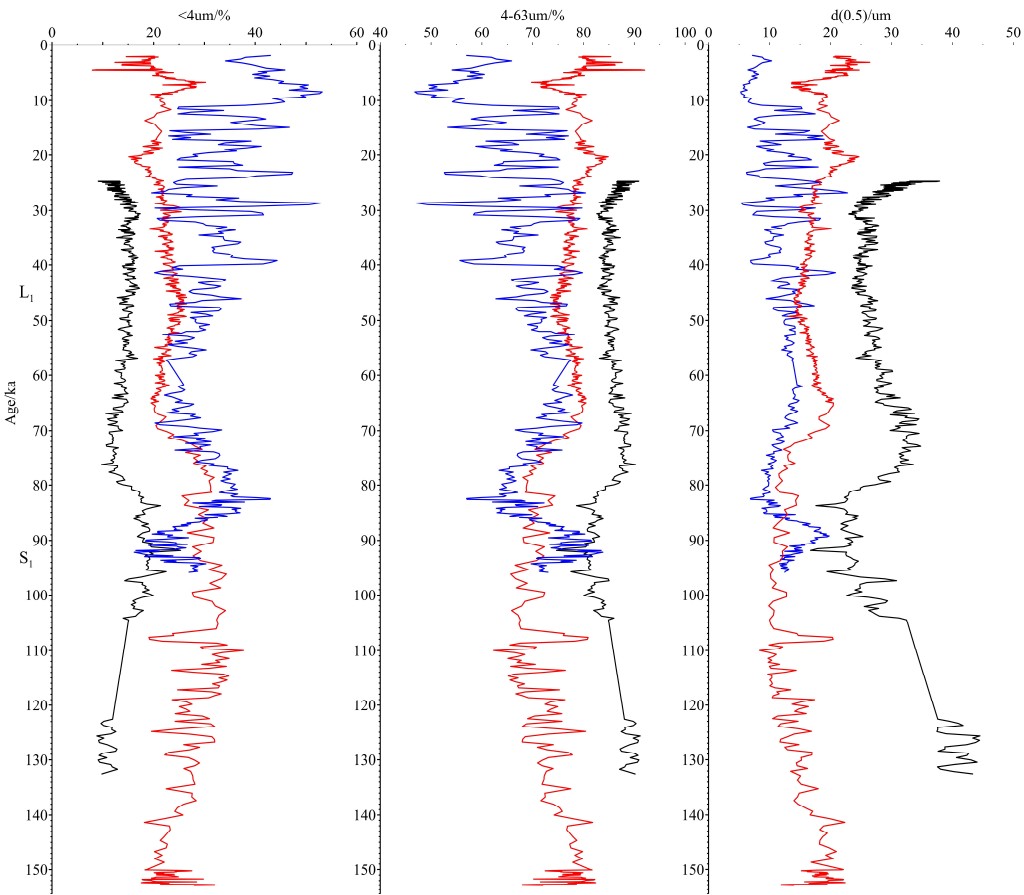

**Figure 4.** Changes in loess grain size characteristics of ZK04, ZK18, and ZK19 since 0.13 Ma. The black, red, and blue curves represent the particle size characteristics of ZK04, ZK18, and ZK19, respectively.

## 5. Discussion

Paleosol is the soil formed on the surface during different geological periods through pedogenesis under the climatic conditions at the time. Therefore, the soil type, composition, structure, and other characteristics of paleosols all show the imprints of the climate characteristics at the time and directly reflect climatic conditions such as temperature and humidity [53,54]. In the northern Luohe Basin, the paleosols were lighter in color and had poorer soil features, whereas those in the southern loess plateau were darker in color and had clear soil features. These characteristics indicated the regional differences in the paleoclimatic conditions in the Luo River Basin during the Quaternary period, i.e., the climate in the north was drier, with less precipitation and lower temperatures compared with the climate in the south.

Most loess researchers have recognized that the alternating appearance of multiple horizons of paleosols and loess in the loess profile reflects changes in the temperature, coldness, dryness, and humidity of the paleoclimate [55–57]. However, changes in the thickness and combination features of the paleosols in the loess profiles indicate that paleoclimatic changes in various regions during the same period often differed. The Last Glacial cycle is the latest complete interglacial and glacial periods in the Quaternary. The change process from the Last Interglacial to the glacial period has important reference value for future climate change of the modern interglacial [58] and, therefore, it has been the focus of research on global climate change in the past. In this study, with climate change since the Last Interglacial as a case study, we investigated the patterns of regional climate change since the Last Interglacial.

Researchers have gleaned ample information about climate change during the Last Interglacial from ice cores, marine sediments, and terrestrial sediments. Although more

comprehensive research has been conducted on the Last Interglacial period, this period also has more dissimilarities. Previous studies [59] have shown that the paleosols of loess profile $S_1$ west of Liupanshan could be divided into three paleosol sublayers; however, some profiles with relatively high sedimentation rates could even be divided into five paleosol sublayers. In the east of Liupanshan, except for some loess profiles with slightly higher deposition rates, most $S_1$ paleosols were found to comprise one or three horizons of paleosols. The current research found that excepting $S_1$ of the JB profile, which could be divided roughly into four paleosol sublayers, $S_1$ of the remaining ZK04, ZK13, ZK18, ZK19, JD, and LC profiles all had only one paleosol horizons (Figure 5). The thickness and deposition rate of $S_1$ gradually decreased from north to south. The thickness of $S_1$ in the JB profile in the north was 4.1 m, and the deposition rate was high at 74.5 cm/ka. The $S_1$ thickness of the central LC profile was 2.3 m, and the deposition rate was 41.8 cm/ka, whereas the $S_1$ thickness of ZK19 in the south was 1.1 m, and the deposition rate was only 20 cm/ka.

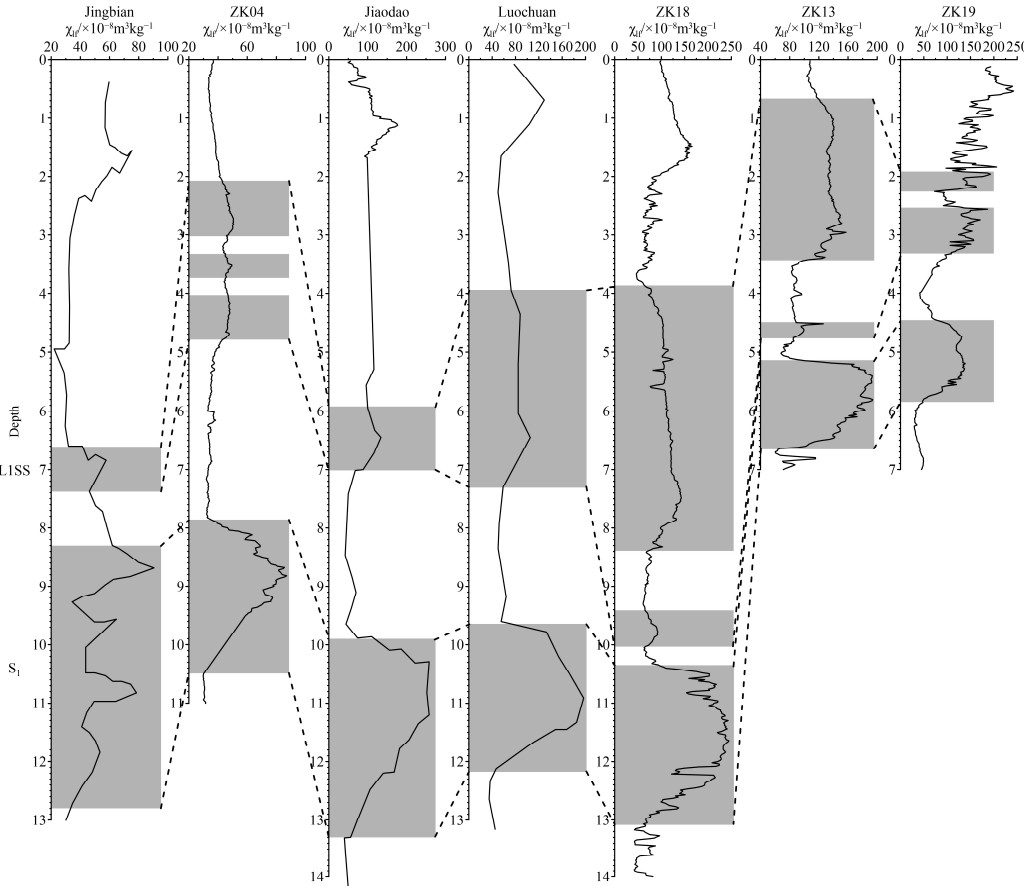

**Figure 5.** Comparison of climate records from representative boreholes and profiles since the Last Interglacial.

The overall climate of $L_1$ during the Last Glacial Period was mainly dry and cold; however, warm and humid fluctuations occurred. A comparison of the $L_1$ changes of four boreholes and three representative profiles in the Luohe River Basin (Figure 5) showed three weak loamy horizons in $L_1$ of ZK04 loess and two weak loamy horizons in $L_1$ of ZK18, ZK13, and ZK19 loess. Owing to the resolution of the samples in the other three representative profiles, only one horizon of weak loamy horizons was found in $L_1$. These findings indicated that climate fluctuations during the Last Glacial Period were inconsistent in different regions.

A degree of regional regularity was reflected by large climatic changes; however, differences occurred in local areas. For example, the lower part of ZK04 comprised lacustrine

sediments, mainly gray-brown clay, gray-green, gray-yellow, or rust-yellow loam and, sometimes, gray-black peat, gray-black silty loam, or sandy loam. These lacustrine sediments contained relatively high carbonaceous components, rich in biochar and sporopollen, and most of the iron elements they contained were in a reduction state with a low degree of oxidation. These characteristics indicated that the climatic conditions in this area at the time were mainly wet and cold. However, the lower part of ZK19 comprised river sediments from coarse lithofacies, such as sand, sand pebbles, and pebble horizons. These coarse lithofacies sediments indicated that a relatively wet period occurred with developed rivers and active hydrodynamic activities when the loess was accumulating, reflecting the humid climate conditions at that time.

## 6. Conclusions

Based on the above analyses, the following conclusions could be drawn.

(1)  The loess thickness and average deposition rate in the Luohe River Basin area varied significantly. The loess thickness ranged from 121 to 252 m, and the average loess deposition rate was between 4.65 and 9.69 cm/ka. In general, both the loess thickness and the loess deposition rate decreased from north to south.

(2)  Paleosol horizons were less developed in the hills and ridges of the northern part of the Luohe River Basin, with lighter paleosols and poorer soil features. In contrast, numerous paleosol horizons occurred in the southern plateau, which were darker in color and showed distinct soil features. The pattern of the degree of paleosol development increased from north to south.

(3)  The magnetic susceptibility variation curves of loess sequences in different regions differed in shape, but the trend was consistent. Both factors showed that the magnetic susceptibility of the loess horizons was low, whereas that of the paleosol horizons was high. For both loess and paleosols, magnetic susceptibility increased from north to south, whereas the particle composition tended to become finer from north to south.

(4)  The climatic fluctuations of the Last Glacial recorded by loess and paleosols in the various regions differed. During the $L_1$ period of the Last Glacial, the overall climate was mainly dry and cold, but the number of fluctuations in temperature and humidity recorded during the period differed. During the $S_1$ period of the Last Interglacial, the climate was mainly warm and humid. The thickness and deposition rates of $S_1$ gradually decreased from north to south, and the recorded paleosol subhorizons also differed.

**Author Contributions:** Methodology, J.S.; validation, J.G., H.C., C.S. and Q.D.; formal analysis, W.W.; investigation, H.C. and C.S.; resources, J.G.; data curation, Q.D.; writing—original draft preparation, J.G.; writing—review and editing, W.W.; visualization, W.W.; supervision, J.S.; project administration, J.G. and W.W.; funding acquisition, J.G. All authors have read and agreed to the published version of the manuscript.

**Funding:** This study was financially supported by the Basic Scientific Research Project (Grant No. SK201403 and SK202217), China Geological Survey Project (Grant No. DD20190433, 1212011120047) and National Natural Science Foundation of China (Grant No. 41877398).

**Data Availability Statement:** The data presented in this study are available on request from the corresponding author.

**Acknowledgments:** We would like to thank Editage (www.editage.cn accessed on 4 April 2023) for English language editing.

**Conflicts of Interest:** The authors declare no conflict of interest.

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
