# Peer review of "Loess Strata Distribution Characteristics and Paleoclimate Spatial Pattern during the Last Interglacial in the Luohe River Basin"

_geosciences, doi:10.3390/geosciences13060158_

Round 1

Reviewer 1 Report

Comments and Suggestions for Authors

General comment:

The respective authors present "Loess Strata Distribution Characteristics and Paleoclimate Spatial Pattern during the Last Interglacial in the Luohe River Basin". Spatial pattern and regional characteristics of the paleoclimate during the Last Interglacial period in the Luohe River Basin, Shaanxi Province, China is addressed by using the magnetic susceptibility and particle size tests. The study is solid and interesting and it does contribute to the field of paleoclimete studies since previous research papers of loess in China have focused mostly on the time series of a single borehole and/or profile. I am stating that minor revision is needed for acceptance of the given paper.    

Specific comments:
- Introduction part (page 1-2; lines 35-68): this section should be enhanced with other studies from the Eurasian perspective (e.g. Comparative analysis of the magnetism between Chinese and Serbian loess deposits; Chinese loess and the Asian monsoon: What we know and what remains unknown; Mineral magnetic properties of loess-paleosol couplets of northern Serbia over the last 1.0 Ma etc.)

- Page 8; lines 276-281: reference is needed to back up this statement.

- Use the term horizons instead of layer/s

- Page 9, lines 302-303: Figure 3. caption is a typo. It should be The characteristics of variations in the low-frequency magnetic susceptibility of the investigated boreholes / classic profiles.

After the above mentioned suggestions for improvement are implemented into the manuscript, I suggest to the Editorial Board of Geosciences scientific journal to consider it for publication. It certainly has great potential considering the research field and methodological approaches.

Kind regards!

I recommend authors to proofread the manuscript before re-submission. Minor corrections are needed.

Reviewer 2 Report

The authors believe that loess and the regularities of its proximity to other sedimentary rocks can contribute to a deeper understanding of the massif hosting minerals when solving issues related to the development of natural resources.

The article presents the results of a study of the spatial orientation, structure and other parameters of rock formation in the basin during the interglacial period. The data of analyzes of loesses and characteristics of the distribution of loesses and products of their contacts in geomorphological regions are presented.

The thickness of the loess, the vector and the rate of its formation in the river basin are revealed. Data on the climatic conditions of glaciation, as well as the features of the distribution of paleosols in the studied regions, are systematized.

The work may be in demand in the exploration and development of mineral deposits.

The disadvantages of the article include the lack of information about the practical applicability of the information obtained.

Based on the relevance of the presented work, the results obtained in it will be useful for the analysis of specialists in this field.

However, there are the following issues that should be clarified:

1. In the article in the review part, a more detailed comparative analysis of the loess strata considered in China with similar strata in other regions of the world should have been given. This is necessary to form a broader understanding of the significance of ongoing research, especially in relation to the analysis of climatic conditions.

2. In Section 3.2, it would be necessary to cite from which literary sources the data on the three classical loess and paleosol profiles were taken.

3. In the article in the "Materials and Methods" section, insufficient attention is paid to research methods. It would be possible to see them on the example of the work: Bosikov I.I., Klyuev R.V., Silaev I.V. Comprehensive analysis and assessment of Prospective gold-ore zones using modern geophysical methods. Geologiya I Geofizika Yuga Russia = Geology and Geophysics of Russian South. (in Russ.). 2022. 12(2): 89-102. DOI: 10.46698/VNC.2022.32.98.007. Also, not enough attention is paid to the scheme of research. It could be shown visually in the form of a figure or given in the form of a table.

4. According to the data presented in tables 1-4, it would be possible to obtain complex mathematical models of the dependence of the desired output values on the parameters under consideration with the corresponding coefficients of determination.

5. Very important data are presented in Figure 3, but the use of these data should be given when testing the results of work in other climatic conditions.

6. In the course of the analysis of Figure 5, it would be necessary, if possible, to provide numerical data on climatic parameters during the last ice age. The article says that they were inconsistent in different regions, a more detailed description should have been provided.

7. In conclusion, we should show the prospects for further application of the results obtained.
